# Diverse Actions of Astrocytes in GABAergic Signaling

**DOI:** 10.3390/ijms20122964

**Published:** 2019-06-18

**Authors:** Masaru Ishibashi, Kiyoshi Egawa, Atsuo Fukuda

**Affiliations:** 1Department of Neurophysiology, Hamamatsu University School of Medicine, Hamamatsu, Shizuoka 431-3192, Japan; masaru@hama-med.ac.jp; 2Pediatric Department, Hokkaido University Graduate School of Medicine, Sapporo 060-8638, Japan; egakiyo@huhp.hokudai.ac.jp; 3Advanced Research Facilities and Services, Preeminent Medical Photonics Education and Research Center, Hamamatsu University School of Medicine, Hamamatsu, Shizuoka 431-3192, Japan

**Keywords:** Astrocyte, GABAergic synapse, Tripartite synapse, GABA, Chloride, Epilepsy

## Abstract

An imbalance of excitatory and inhibitory neurotransmission leading to over excitation plays a crucial role in generating seizures, while enhancing GABAergic mechanisms are critical in terminating seizures. In recent years, it has been reported in many studies that astrocytes are deeply involved in synaptic transmission. Astrocytes form a critical component of the “tripartite” synapses by wrapping around the pre- and post-synaptic elements. From this location, astrocytes are known to greatly influence the dynamics of ions and transmitters in the synaptic cleft. Despite recent extensive research on excitatory tripartite synapses, inhibitory tripartite synapses have received less attention, even though they influence inhibitory synaptic transmission by affecting chloride and GABA concentration dynamics. In this review, we will discuss the diverse actions of astrocytic chloride and GABA homeostasis at GABAergic tripartite synapses. We will then consider the pathophysiological impacts of disturbed GABA homeostasis at the tripartite synapse.

## 1. Introduction

Glial cells were initially thought to be stroma that only filled in between neurons in the central nervous system. Subsequent research shows that glial cells not only support nerve cells structurally, but also provide various physiological functions such as supplying nutrients, homeostasis of various substances around synapses, and neuroprotection [1]. Furthermore, the ratio of glial cells to neurons in the mammalian brain increases with brain size, suggesting that there may be some correlation between brain evolution and the number of glial cells [2].

At present, glial cells are classified into astrocytes, oligodendrocytes and microglia, and are identified by specific antigen molecules. Among these, astrocytes, which will be described in detail in this review, have been shown to be involved in neuronal signal transduction, and have been actively studied in recent years. Astrocytes extend fine processes in all directions that are structurally supported by the intermediate filament, glial fibrillary acidic protein (GFAP). This GFAP is widely used as a marker for identifying astrocytes [1]. The processes are branched from stem process into lamellae, and then further branched into the microprocesses of filopodia, which are embedded between nerve cells. The astrocytic filopodia are very thin and closely wrap synapses, forming a structure now called the “tripartite synapse” [3]. In addition to these anatomical and structural features, the ability of astrocytes to release glutamate and GABA following monoamine stimulation was known prior to the discovery of elevated calcium levels in astrocytes [4,5], although the release mechanism remains unknown. Subsequent studies have shown that astrocytes can release neurotransmitters, including glutamate, ATP, GABA, and glycine by various mechanisms [1]. However, the effect of astrocytes on synaptic transmission at tripartite synapses has mostly focused on excitatory synaptic transmission by glutamate and ATP [6], although the pathway by which glutamate from astrocytes is synthesized into GABA by GAD in inhibitory neurons is widely recognized [7,8]. In recent years, substantial new evidence has emerged indicating a role of astrocytes in regulating GABAergic neurotransmission. In this review, we will first discuss GABA-mediated ionotropic signaling from neurons to astrocytes. We will then discuss GABA transport systems in astrocytes. We will then finally discuss the pathophysiological impact that is thought to result from disturbance in GABA homeostasis by astrocytes.

## 2. GABA-Mediated Signaling from Neurons to Astrocytes

### 2.1. GABA_A_ Receptor-Mediated Neuron–Astrocyte Signaling

GABA_A_ receptors are assembled from 19 subunits (six α subunits, three β subunits, three γ subunits, three ρ subunits, and one each of the ε, δ, θ, and Π subunits), forming mostly hetero-oligomeric pentamers [9]. While brain region or neuronal subtype-specific patterns of subunit expression have been deeply explored in neurons, the subunit expression pattern of astrocytic GABA_A_ receptors is still largely unknown. Single-cell RT-PCR, which was recently performed for analyzing GABA_A_ receptor α, β, and γ subunit expression, showed intense expression of γ1 and γ3, but lack of γ2 subunit in astrocytes [10], in accordance with findings from a previous electron microscopy study in Bergmann glia of the cerebellum using postembedding immunogold techniques [11]. A transcription study comparing freshly isolated neurons and astrocytes also showed exclusive expression of γ1 in astrocytes and γ2 in neurons [12,13]. Because γ2 is the most predominant component among the three γ subunits and is crucial for GABA_A_ receptor trafficking to the synaptic cleft [14], the lack of γ2 subunit in astrocytes may suggest a non-synaptic form of GABA_A_ receptor-mediated activation in those cells. Besides the major α, β, and γ subunits, the ρ subunit is expressed in astrocytes of various brain regions including the cerebellum [15,16] and the neostriatum [17] during development. GABA_A_ receptors containing the ρ subunit show bicuculline-insensitive, non-desensitizing activation by GABA [17] and are thought to play a role in glial migration and differentiation during development, although their functions are still under investigation.

The term “tripartite synapse” underscores the important participation of astrocytes in central nervous system information processing mediated by chemical synaptic transmission. This concept is largely attributed to the structural elucidation of astrocytic processes tightly wrapping neuronal synapses and expressing receptors [18] as well as transporters [19] for neurotransmitters. In terms of ionotropic GABA-mediated signaling, a number of pioneering works have demonstrated that astrocytes express both GABA_A_ receptors and transporters and that they sense exogenously applied GABA or GABA_A_ receptor agonists [20]. However, direct evidence for a structural junction between GABAergic synapses and astrocytic processes has been limited to the NG2 glia in the hippocampal stratum radiatum (SR) region [21] or to the Bergmann glia, whose processes wrap parallel fiber input to Purkinje cells in the cerebellum [11]. Further, a limited study has evaluated trafficking and localization of receptors or transporters for GABAergic signaling on astrocytes [11]. Thus, the degree to which astrocytes can contribute to the modification of synaptic GABA-mediated signaling is largely unclear. We previously compared astrocytic GABA-mediated currents resulting from the firing of a single interneuron with those resulting from exogenous GABA application in the hippocampus. While exogenous GABA induced GABA transporters (GATs), as well as GABA_A_ receptor-mediated currents, a train of action potentials in a single interneuron exclusively induced GABA_A_ receptor-mediated currents [22]. Although it is generally assumed that astrocytes can sense spillover GABA from the synaptic cleft and modulate GABAergic signaling in an extrasynaptic manner, in line with the lack of a postsynaptic γ2 subunit in astrocytes as mentioned above, this result suggests that astrocytic GABA_A_ receptors are localized closer to the synaptic clefts than GATs. This insight is supported by an electron microscopy analysis showing that Bergmann glia GABA_A_ receptors are often located in close proximity to GABAergic synapses (type II, symmetric) on Purkinje cells [11]. Thus, astrocytes may be able to sense the release of GABA from synaptic terminals via GABA_A_ receptors. 

Contrary to the “tripartite synapse” theory, a type of interneuron named neurogliaform cell does not form conventional synapses. Instead, it releases GABA and activates receptors at a distance via volume transmission [23]. Astrocytic GABA_A_ receptor currents can also be activated by this volume transmission and can be detected even following a single spike in a neurogliaform cell [24]. The functional role of this type of neuron–astrocyte signaling is still unclear but may contribute to the regulation of the extrasynaptic GABA_A_ receptor-mediated tonic inhibition of neurons. 

In contrast to neurons, the GABA_A_ receptor activation of astrocytes causes membrane depolarization in cell culture [20,25] and in situ [26,27] throughout postnatal development. Because the GABA_A_ receptor is a ligand-gated chloride (Cl^−^) channel, this GABA-mediated depolarization can be simply interpreted as Cl^−^ efflux resulting from a negative driving force due to high intracellular Cl^−^ concentration. Indeed, by using gramicidin perforated patch-clamp recording, a higher [Cl^−^]i value (20–40 mM) has been shown in astrocytes in culture [27], which is presumably due to the activity of the Cl^−^ importer Na^+^/K^+^/2Cl^−^ co-transporter (NKCC1) [28]. In studies using cultured astrocytes, this GABA_A_ receptor-mediated depolarization may function to cause a rise of calcium (Ca^2+^) [29,30,31]. Nevertheless, there has been no evidence showing a rise of astrocytic Ca^2+^ via GABA_A_ receptor activated by endogenously released GABA from presynaptic neurons. Taking into account the extremely low membrane input resistance of mature astrocytes, it would be difficult for currents produced by endogenously released GABA to activate astrocyte voltage-sensitive Ca^2+^ channels (VOCCs).

Alternatively, Kettenmann et al. hypothesized that this Cl^−^ efflux from astrocytes could buffer the [Cl^−^]o of the GABAergic synapse and contribute to maintaining GABAergic neuronal transmission [20]. The appeal of this hypothesis was increased by follow-up evidence showing that the neuronal [Cl^−^]o/[Cl^−^]i gradient can collapse due to intracellular Cl^−^ accumulation following GABA_A_ receptor activation [32] and our previous finding that GABA_A_ receptor-mediated astrocytic currents propagate via gap junctions within the astrocytic network. Indeed, we found that the pharmacological inhibition of gap junctions by octanol enhanced the collapse of the neuronal [Cl^−^]o/[Cl^−^]i gradient as represented by depolarizing shifts of the neuronal IPSC reversal potential evoked by repetitive stimulation to GABAergic synapses [22] (Figure 1A). Thus, the astrocytic network connected by gap junctions may contribute to maintaining GABAergic transmission by spatial buffering of [Cl^−^]o during the intense activation of neuronal networks. It has been generally assumed that [Cl^−^]o is much higher and more stable than [Cl^−^]i so that, according to the Nernst equation, its change has limited effect on modulating reversal potential of Cl^−^. However, recent studies illustrated that [Cl^−^]o can be variable depending on the amount of membrane impermeable anions in the extracellular matrix, which determine [Cl^−^]i in accordance with the Donnan effect [33]. [Cl^−^]o in the micro-domain might be much more dynamic than previously assumed [34]. Thus, astrocytes may act to store Cl^−^, and the activation of their GABA_A_ receptors could help maintain [Cl^−^]o at the synaptic cleft. 

On the other hand, the notion of a higher [Cl^−^]i in astrocytes is currently under debate. Gramicidin perforated-patch clamp recordings in acute hippocampal slices showed that the equilibrium potential of astrocytes was close to that of the resting membrane [35], which means that no Cl^−^ current is evoked by GABA_A_ receptor activation in the resting condition [36]. Alternatively, the efflux of HCO_3_^−^ and/or inhibition of inwardly rectifier potassium channels were proposed as a mechanism(s) for GABA_A_ receptor-mediated depolarization in astrocytes [35]. However, as the same group demonstrated elegantly using dual whole cell patch-clamp recording [37], extremely low input resistance of mature astrocytes causes large voltage clamp errors, especially under the high access resistance conditions of gramicidin perforated patch-clamp recordings. Therefore, other approaches, such as Cl^−^-imaging, are needed to evaluate [Cl^−^]i in astrocytes. In summary, a functional role of astrocytic GABA_A_ receptor-mediated signaling remains to be fully elucidated. Further evaluation, for example, using astrocyte-specific GABA_A_ receptor knockout mice might be required to address this issue.

### 2.2. GABA_B_ Receptor-Mediated Neuron–Astrocyte Signaling

GABA_B_ receptors are Gi/o type G-protein-coupled, metabotropic receptors that induce slow inhibitory signaling in both presynaptic and postsynaptic neurons. The activation of the presynaptic GABA_B_ receptor reduces transmitter release by inhibiting recruitment of neurotransmitter vesicles to the cell membrane or by inhibiting voltage-gated calcium channels. Postsynaptic GABA_B_ receptors induce slow hyperpolarization by activating inwardly rectifying potassium channels [38]. Astrocytes also express GABA_B_ receptors, and early studies showed that their activation by GABA_B_ receptor agonists (Baclofen) [30,39,40] or by synaptically-evoked GABA release [40] induced the elevation of astrocytic Ca^2+^. This GABA_B_ receptor-mediated astrocytic Ca^2+^ transient can modify synaptic transmission efficiency, as shown by the presynaptic metabotropic glutamate receptor (mGluRs)-mediated transient reduction of glutamate release (known as transient heterosynaptic depression) [39] or the potentiation of miniature inhibitory postsynaptic currents in neighboring pyramidal neurons [40]. 

Recent publications using modern genetic and/or optical techniques have provided cumulative evidence for the GABA_B_ receptor-mediated astrocytic Ca^2+^ transient and its effects on synaptic processing. In vivo, Ca^2+^ imaging utilizing two-photon microscopy has indicated that astrocytic GABA_B_ receptor activation triggers a long-lasting Ca^2+^ oscillation in the cortex of living brain, which can evoke glutamate release as a gliotransmitter and induce slow glutamatergic currents in neighboring pyramidal neurons [31]. Similar Ca^2+^ oscillations and consequent glutamate release were replicated by applying a pathophysiologically high potassium solution [41]. The effects of the astrocytic Ca^2+^ oscillations on synaptic transmission have been demonstrated by its blockage with 1,2-bis(o-aminophenoxy)ethane-*N*,*N*,*N*′,*N*′-tetraacetic acid (BAPTA, Ca^2+^ chelator) dialyzed into the gap junction-coupled astrocytic syncytium [40,41,42] or with knocking out *IP3R2* [31,41,42], a subtype of IP3 receptors dominantly expressed in astrocytes. More specifically, Perea et al. generated conditional astrocyte-specific GABA_B_ receptor knockout mice (GB1-cKO mice) and showed that interneuron spike trains evoked astrocytic Ca^2+^ oscillation mediated by astrocytic GABA_B_ receptors, which triggered potentiation of excitatory synaptic transmission via the activation of presynaptic metabotropic glutamate receptors [42]. Notably, hippocampal local field potential recordings in GB1-cKO mice revealed reduced stimulus-driven theta and low gamma band activities, which indicates the impact of astrocytic GABA_B_ receptor activation on activity-dependent neuronal oscillations.

In contrast to the excitatory pyramidal neurons, inhibitory interneurons are highly heterogeneous. For example, parvalbumin (PV)-positive fast-spiking basket cells make perisomatic and proximal dendritic terminals in contrast to somatostatin (SST)-positive interneurons, which show low threshold spiking and target dendritic tufts [43]. This diversity is one of the reasons why the properties of GABAergic tripartite synapses remain to be elucidated. Two recent studies utilizing a cell type-specific optogenetic technique have addressed this issue. In the somatosensory cortex [44] or the hippocampus [45], astrocytic GABA_B_ receptor-mediated Ca^2+^ transients were dominantly [44], or exclusively [45], generated by firing in SST-positive interneurons rather than PV-positive interneurons. The precise mechanism is still unknown; however, Mariotti et al. illustrated that SST release from SST-positive interneurons was required to facilitate the GABA_B_ receptor-mediated Ca^2+^ oscillation [44]. Although the effects of other neuropeptides (i.e., neuropeptide Y or cholecystokinin) synthesized by distinct interneuron classes still remain to be elucidated, these findings suggest that astrocytes can sense distinct interneuron signaling and affect the tuning of neuronal computational processing by potentiating the presynaptic function of a specific type of interneuron(s). In this view, the subsequent propagation of GABA-mediated astrocytic Ca^2+^ transient to neighboring astrocytes and its functional role are of further interest. If the wave-like propagation of [Ca^2+^]i increases via gap junctions, or ATP release [46] is mediated by GABAergic signaling, astrocytes could spatially code the computational function of distinct interneurons.

While cumulative evidence has confirmed its occurrence, the precise intracellular mechanisms mediating Ca^2+^ transients via astrocytic GABA_B_ receptor activation is still unclear. From data using the IP3R2 knockout mice mentioned above, or pharmacological blockage [42], the elevated Ca^2+^ should be released from the internal Ca^2+^ stores. As it is generally assumed that Ca^2+^ release from the internal store requires Gq protein activation, this result raises the question of why Gi/o type G-protein coupled GABA_B_ receptors can trigger the Ca^2+^ release [7]. One possible explanation is the interaction between GABA_B_ receptors and other Gq type metabotropic receptors. In cultured cortical astrocytes, the pre-activation of P2 purinoceptors (P2YRs) was shown to be required for induction of GABA_B_ receptor-mediated Ca^2+^ elevation [47]. Alternatively, the augmentation of co-expressed mGluR1 by GABA_B_ receptor activation shown in Purkinje cells [48] might be applicable in astrocytes [30]. However, these speculations are still controversial as a blocker for P2YRs [30,41] or mGluRs [41,42] did not reduce astrocytic GABA_B_ receptor-mediated Ca^2+^ transient. Thus, further research is required to address the intracellular mechanisms regarding GABA_B_ receptors. 

### 2.3. GABA Transporter-Mediated Neuron–Astrocyte Signaling

GABA transporters (GATs) expressed in astrocytes not only play an important role in regulating the ambient GABA concentration in the extrasynaptic space (for details, see Section 3), but can also directly participate in the GABA-mediated signal transduction pathway within a tripartite synapse. Several studies have shown that GABA-mediated Ca^2+^ transient is prevented by GAT3 inhibitors [45,49,50]. Because GAT3 (also known as, mouse GAT4) is assumed to be expressed exclusively in astrocytes, these pharmacological experiments indicate that GAT activation by GABA results in Ca^2+^ elevation in astrocytes. The mechanism of this is as follows: an increase of intracellular Na^+^ has been shown as a consequence of co-transporting Na^+^ with GABA via GATs [49,50]. This can induce Ca^2+^ rise by inhibiting Ca^2+^ efflux via a Na^+^/Ca^2+^ exchanger [49] or by even facilitating Ca^2+^ influx via a Na^+^/Ca^2+^ exchanger in reverse mode [50]. This astrocytic Ca^2+^ elevation triggers ATP release, which diffusely activates presynaptic adenosine receptors and decreases glutamate release from excitatory synaptic terminals [50] or activates postsynaptic adenosine receptors and potentiates GABA-mediated postsynaptic inhibitory currents [45]. 

Evidence of astrocytic GAT involvement in neuron–glia signal transmission is limited to data regarding their pharmacological blockage. It is also still unclear whether this signal transduction pathway can occur under physiological conditions. These issues need to be addressed by further analysis using conditional knockout mice to eliminate astrocyte-specific GATs.

## 3. GABA Transport Systems in Astrocytes

### 3.1. GABA Transporter (GAT)

In general, it has been thought that inhibitory signal transmission is mediated primarily by chemical synaptic transmission, where GABA is released into the synaptic cleft and acts on postsynaptic receptors. This GABA action is then rapidly terminated by uptake into presynaptic, postsynaptic and/or astrocytic elements by a specific and high affinity GABA transporter expressed in perisynaptic membranes [51,52]. This GABA transporter (GAT) is a 12-transmembrane protein, and belongs to the Na^+^ and Cl^−^ -dependent transporter family (soluble carrier superfamily (SLC) six gene family), which also includes the monoamine (dopamine, serotonin and noradrenaline), glycine, amino acid, creatine and osmolyte (betaine and taurine) transporters [53,54]. To date, four subtypes of GAT have been identified from mammalian tissues. In mice, the four subtypes are named GAT1, GAT2, GAT3 and GAT4 [55], which are sometimes written mGAT1 to mGAT4. However, in humans, the same gene products can have different names (see Table 1) [56]. It is, therefore, important to indicate the species when referring to particular GAT subtypes. In this review, the notation of GAT subtypes refers to the human subtype names.

GAT is an electrogenic GABA-selective transporter that requires cotransport of two Na^+^ ions and one Cl^−^ ion for uptake of one molecule of GABA, and it is thought that it can also be driven in reverse mode. The reversal potential of GAT is considered to be approximately −50 mV [1]. By patch-clamp experiments using brain slice preparations, it has been reported that cerebellum Bergmann glia show an electrogenic GABA efflux that activates nearby GABA_A_ receptors, and it has been shown that GABA can be released from the glia cell body by reverse-mode operated GAT [57]. In addition, GABA transporters expressed at synaptic terminals of neurons and astrocytes around synapses are involved in the regulation of the synaptic cleft and extrasynaptic GABA concentration and influence the amplitude of tonic and phasic synaptic GABA currents in the hippocampal dentate gyrus [58]. From these results, GAT not only takes up GABA that has overflowed the synaptic cleft, but also releases or buffers GABA in the perisynaptic region depending on the concentration gradients of Na^+^, Cl^−^ ions and GABA. Thus, GAT is thought to play an important role in GABA concentration homeostasis in the perisynaptic extracellular space of the central nervous system.

Among the GABA transporters identified so far, most are mainly expressed in neurons although GAT3 is thought to be predominantly expressed in astrocytes [51,59]. For example, in the rat and human neocortex, GAT1 and GAT3 are the main isoforms expressed, with GAT1 mainly in neurons [60,61], and GAT3 mainly in rat neocortical perisynaptic astrocytic processes [62]. GABA transport in rat astrocytes was also shown to be suppressed in a concentration-dependent manner by Zn^2+^, which has been reported to have strong inhibitory effects on GAT3 [63]. It has also been reported that BGT1 (GAT2 in mouse) mRNA is expressed in cultured astrocytes and astrocytoma cell lines, but its expression has not been confirmed in cultured neurons. In addition, BGT1 protein has been reported in astrocytes and astrocytoma cells cultured in a hyperosmotic environment. Evidence from rat primary astrocyte cultures support this, since GAT2 as well as GAT3 are expressed and function in astrocytes [63]. However, as the specificity of the rat BGT1 antibody used for these studies has been questioned, further verification may be needed for the expression of BGT1 (GAT2 in mouse) in astrocytes [64]. GAT3 has been reported to be expressed at high levels on Bergmann glia around perisynaptic region in the cerebellum. Since GAT is not expressed in Purkinje cells, GABA released into the synaptic cleft of these inhibitory synapses is considered to be taken up by GAT3 expressed around these synapses. Furthermore, it has been reported by immunocytochemical localization that GAT3 is expressed at high levels in astroglial processes near symmetric and asymmetric synapses [65]. In rat hippocampal CA3 and dentate gyrus, GAT1 and GAT3 are expressed, with GAT3 expressed in astrocytes, where it is involved in regulation of the perisynaptic GABA concentration [66]. This perisynaptic GABA activates extrasynaptic GABA_A_ receptors and induces a persistent noisy current in neurons called the tonic GABA current. 

It has also been shown that the expression of GAT3 is dependent on intracellular calcium concentration in astrocytes and that the expression level of GAT3 precisely regulates the efficiency of inhibitory synaptic transmission in co-culture preparations of hippocampal neurons and astrocytes [67]. In this study, the authors showed that transient receptor potential A1 (TRPA1) channels mediate frequent and highly localized Ca^2+^ transients in astrocytes near the membrane that contributed to an overall rise in [Ca^2+^]i. The inhibition of TRPA1 channel reduced GAT3 membrane expression by reducing [Ca^2+^]i. This study underscores the importance of astrocytes in finely influencing inhibitory synaptic transmission since decreasing GABA transport by GAT resulted in an increase in the extracellular GABA concentration and a desensitization of the GABA_A_ receptor leading to reduced efficiency of inhibitory transmission [67].

GAT expression in the thalamus is different from other brain regions such as the neocortex and hippocampus, and although GAT3 expression is strong in astrocytes, GAT1 is also selectively expressed in astrocytes [68], and expression is abundant postnatally [69]. Furthermore, GAT1 is involved with the enhancement of the tonic GABA current in thalamocortical neurons in a rodent model of absence epilepsy [70,71]. It has also been shown that GAT1 and GAT3 affect the dynamics of GABA_B_ receptor-dependent postsynaptic currents, and GAT1 is expressed near the synapse and affects the maximum amplitude of IPSC, whereas GAT3 is widely expressed in distal extrasynaptic regions and limits diffusion of GABA from the synapse [72].

Calcium-dependent GABA release from astrocytes possibly involving GAT has also been reported in the dorsal root ganglia of red-eared turtles [73]. This GABA release is caused by the puff application of glutamate and is not affected by the removal of extracellular calcium or extracellular addition of cadmium, nickel or tetrodotoxin, whereas intracellular calcium chelation prevents release of GABA. In addition, intracellular calcium elevation is observed in astrocytes of the dorsal root ganglion by selective stimulation of Aδ and C fibres, suggesting that this GABA release may be involved in the integration of sensory inputs. However, although GAT3 expression has been confirmed in astrocytes, the Ca^2+^-dependent release mechanism remains unclear [73]. In addition to glutamate, the modulation of GABA transport by the activation of adenosine A1–A2 heteromeric receptors has been reported. This regulation occurs through A1R–A2_A_R heteromers that signal via two different G proteins, Gs and Gi/o, and either enhances (A2_A_R) or inhibits (A1R) GABA uptake [74]. Also, besides direct effects on synaptic transmission, it has been reported that reduction of astrocyte-expressed GAT causes a reduction in the number of inhibitory synapses in drosophila [75]. Taken together, these findings indicate that the modulation of the extracellular GABA concentration by the reuptake or release of GABA from astrocytes may affect synaptic transmission through various pathways.

### 3.2. Other Mechanisms for GABA Release from Astrocytes: Anion Channels and Gap Junction Hemichannels

It has been suggested that the activation of neuronal GABA receptors can also occur by GABA permeating through Bestrophin (Best) anion channels at the tripartite synapse. To date, four members of the Best family have been identified (Best-1 to 4) encoded by four *BEST* genes (*BEST1*, *BEST2*, *BEST3* and *BEST4*) in humans. It has been suggested that the Bestrophin channel is assembled from multimers of at least dimers or more, and that each subunit has four transmembrane regions. Moreover, the most prominent feature is that the Bestrophin channel is an intracellular Ca^2+^-dependent chloride channel [76,77]. The expression of Best-1 has been widely reported in both nerves and glia, and Best-2, in particular, has been observed in olfactory nerves. It has been suggested that large molecules such as amino acid neurotransmitters may also be conducted, but in recent years, it has been reported that synaptic transmission and neuronal excitability are regulated by the release of glutamate [78] and GABA [79] from the astrocyte through the Best-1 channel. Studies of Best-1 permeability to glutamate and GABA, show that GABA has a lower permeability than glutamate [79]. “Sniffer-patch” experiments [80,81] have shown that the Best-1-mediated release of glutamate and GABA is dependent on intracellular calcium and is triggered by G-protein coupled receptor activation [78,79,82]. Moreover, it has been shown that release occurs at intracellular calcium concentrations close to resting levels, suggesting that other stimuli that may elevate intracellular calcium can cause release [79]. These experiments also suggest that GABA released from astrocytes produces a tonic GABA current in cerebellar granule cells, since astrocyte-selective silencing of Best-1 with shRNA reduced this current [79]. In the hippocampal CA1 region, the tonic GABA current recorded in pyramidal cells is smaller than that in cerebellar granule cells despite Best-1 being expressed in nearby astrocytes. This may be related to the much lower amount of GABA in these astrocytes since current amplitude should depend on the GABA gradient [83]. Thus, GABA release may not occur under physiological conditions in CA1. 

Recently, it has been reported that the tonic GABA current in cerebellar granule cells is reduced by the genetic or pharmacological suppression of Best-1 channels, which increases the excitability of cerebellar granule cells, enhances synaptic transmission between parallel fibers and Purkinje cells and modulates motor function [84]. In contrast, it has been reported that pharmacological inhibition by NPPB, an inhibitor of calcium-sensitive chloride currents, increases the tonic GABA current in cerebellar granule cells [85]. This result is different from that of Lee et al. [79]. Yoon et al. [86] argues that the tonic GABA current was suppressed after 10 min because they have recorded the early enhancement part of the biphasic effect of NPPB on tonic GABA current. However, there is controversy regarding these two different results. Apart from these studies, it has been reported that the administration of NPPB also increases the amplitude of the tonic GABA current in the hippocampus [87]. This result suggests that the function of Best-1 is buffering GABA from inhibitory synapse rather than releasing GABA by diffusion through those channels in the hippocampus. Therefore, the function regarding GABA homeostasis by the Best-1 is controversial.

In another study, it has been suggested that volume-regulated anion channels (VRAC) are involved in the diffusive release of GABA from astrocytes [88]. VRAC, also called the cell volume-sensitive outward rectifying anion channel (VSOR), is a chloride channel that opens with increasing cell volume in many cell types, including glia. VRAC has also been reported to be permeable to the free amino acid taurine [89]. Since taurine is a partial agonist of the GABA_A_ receptor, it is possible that taurine produces the same kind of action as GABA diffusion from astrocytes. However, no clear evidence for this hypothesis has been reported.

In addition to GATs and anion channels, the modulation of the extracellular GABA concentration via the gap junction hemichannel on astrocyte has been suggested. Astrocytes are known to be coupled to nearby astrocytes by gap junctions thereby forming extended networks [90]. A gap junction is formed by the alignment of connexons expressed on each cell membrane, which brings the two cells into a connected state. Connexons are assembled from six subunits of connexin in each cell to form a connexin hemichannel (also termed a gap junction hemichannel). So far, 21 connexin (Cx) isoforms have been identified in humans and 20 isoforms in mice. They are typically named by molecular weight, so a Cx of 40 kDa molecular weight is termed Cx40 [91]. Cx30 and Cx43 are expressed mainly in astrocytes [92,93,94]. The pore of the gap junction hemichannel is permeable to ions and small molecules of 1 kDa or less, including second messengers and various amino acids [95]. In addition to forming gap junctions, hemichannels also exist unpaired in intracellular membranes and the plasma membrane. It has been reported that those unpaired hemichannels release or reuptake neurotransmitters such as glutamate and ATP, depending on the concentration gradients of those neurotransmitters [96,97]. Despite the possibility, evidence for GABA permeability has only recently been reported, indicating that GABA release via gap junction hemichannels is involved in the regulation of tonic GABA currents of neurons in culture and acute hippocampal slices [87]. Thus, gap junction hemichannels appears to be another route by which GABA can be released from astrocytes to influence inhibitory synaptic transmission or extracellular GABA concentration homeostasis.

## 4. GABA Synthesis in the Astrocytes

GABA within astrocytes has been thought to be due to the uptake of GABA released from inhibitory synapses by the GAT expressed on astrocytes. However, considering that cytoplasmic GABA is rapidly metabolized by GABA transaminase, uptake alone may only produce a low GABA concentration in the astrocyte cytoplasm [98]. It has, therefore, been suggested that the reuptake of ambient GABA alone is insufficient for astrocytes to release GABA by diffusion through anion channels or the reverse mode operation of GAT. There may be a GABA supply pathway other than reuptake. In support of this hypothesis, it has been reported that astrocytes, in various sites such as the brainstem [99], hippocampus [100], cerebellum [79,101], neonatal optic nerve [102,103] and human white matter [104] not only contain GABA but also synthesize GABA via GABA synthetase or glutamic acid decarboxylase (GAD67 or GAD65). In addition to GABA synthesis by these GADs, involvement of the GABA synthesis pathway from the polyamine putrescine has also been reported [52,105]. Recently, it has also been shown that monoamine oxidase B (MAOB) plays an important role in GABA synthesis from putrescine in astrocytes [106]. For example, it has been reported that the genetic modification of astrocyte MAOB expression affects extracellular GABA levels [84]. Consistent with this notion, conditioned medium from cultured hippocampal glial cells contained up to 500 µM GABA, whereas almost no GABA was detected in conditioned medium from cultured hippocampal neurons. In addition, it has been reported that the amount of GABA released into the conditioned medium of cultured hippocampal glial cells is increased by the addition of glutamate, which is a precursor for GABA synthesis [107]. It has also been reported by studies using immunohistochemistry that the amount of GABA detected in astrocytes in these various brain regions varies considerably. For example, the astrocytic GABA content in the cerebellum is higher than it in the hippocampus [83,86]. In addition to the pathway through MAOB in the GABA production from putrescine, the production pathway through diamine oxidase (DAO) is also evident in astrocytes [108,109]. Despite this evidence, whether sufficient GABA is synthesized for release by astrocytes remains controversial. 

This question is further complicated by the possibility that astrocytic GABA synthesis varies across developmental stage. For example, while the expression of GABA and GAD in optic nerve astrocytes has been confirmed, the expression is transient during development [103]. In addition, induction of the tonic GABA current in mature cerebellar granule cells no longer depends on the firing of surrounding neurons [110]. The ability for astrocytic GABA synthesis may also depend on the state of the astrocyte. Glutamate released extracellularly from excitatory synapses is taken up by astrocytes via EAAT1 (GLAST) and EAAT 2 (GLT-1) transporters, and then glutamate is converted to glutamine by the glutamine synthetase (GS). Since this glutamine is used as a substrate for GABA synthesis, it has been suggested that GS dysfunction in reactive astrocytes may cause a deficiency in GABA production [111]. In ischemia experiments using adult gerbils, it has been reported that in various regions damaged by ischemia, such as the hippocampus, striatum, and forebrain including layers 2/3 of the somatosensory and auditory cortices, there is an increased number of GFAP-positive cells that are double-labeled for GABA, but not for GAD. Those result suggested that these cells may synthesize GABA via putrescine. In addition, those changes in damaged tissue are observed even after three months [112]. Taken together, GABA release from astrocytes involves not only GABA reuptake from the extracellular space by GAT and Best-1, but also GABA synthesis within astrocytes via multiple pathways, which may change according to developmental stage and pathophysiology conditions.

## 5. Pathophysiological Impacts

The reports on the morphological changes of glial cells in various diseases date back to the 19th century. Since then, studies have suggested a relationship between glial dysfunction and various neurological diseases such as epilepsy, Alzheimer’s disease, degeneration after stroke or focal cerebral ischemia and amyotrophic lateral sclerosis [6]. Moreover, altered astrocytes have also been reported in the preclinical stage of these diseases, suggesting that astrocytes might play a role in their underlying pathogenesis. Therefore, in this section, we have focused on the relationship between altered astrocytes and GABA homeostasis in these diseases. 

Epilepsy is a neurological disease that causes abnormal excess synchronous neural activity, and various mechanisms are involved in epileptogenesis [8]. In the experimental absence epilepsy animal model, it has been reported that the amplitude of the tonic GABA current recorded in thalamocortical neurons is enhanced, resulting from an increase in ambient GABA levels caused dysfunction of GAT1 expressed in ventrobasal thalamus astrocytes [70,71]. It has been suggested that the enhancement of this tonic GABA current causes excessive suppression of thalamocortical neurons and reduction of integrated input to cortical neurons.

Astrocytes also appear to be involved in Parkinson’s disease, a progressive neurodegenerative disease resulting from the loss of dopamine neurons with resting tremor, difficulty in initiating movement and a shuffling gait. In dopamine-depleted animals, it has been reported that astrocytic GAT3 expression is down-regulated, which causes synaptically released GABA to diffuse out of the synaptic cleft into the extrasynaptic space. This increases extrasynaptic GABA_A_ receptor activity of inhibitory transmission in the external globus pallidus and causes motor signaling disruption [113].

In Alzheimer’s disease (AD) symptoms of cognitive decline and personality change result from neuronal cell death, in part, related to accumulation of amyloid-beta (Aβ). In cultured astrocyte experiments, it has been reported that Aβ25-35, an active fragment of Aβ, activates astrocyte gap junction hemichannels to release glutamate and ATP, which damages surrounding neurons [114]. In human preclinical, end-stage AD and APP/PS1 AD model transgenic mice [115], it has been reported that the excitatory vesicular glutamate transporter 1 (vGlut1) was reduced while the density of the vesicular GABA transporter (vGAT) at presynaptic terminals was unchanged [116]. In contrast, it has been reported that GAD activity is specifically elevated in glia of the cerebellum and cerebral cortex of APP/PS1 transgenic mice where the accumulation of Aβ plaques occur [116]. These results suggest that Aβ plaques stimulate GABA production in astrocytes. Furthermore, it has been shown that reactive astrocytes produce GABA in abnormally large amounts due to the abnormal activity of MAOB and release abnormal amounts of GABA via Best-1 channels [82]. In the dentate gyrus of an AD model mouse, abnormally released GABA activates presynaptic GABA_A_ receptors and reduces the firing rate of granule cells [82]. The inhibition of GABA production in astrocyte or GABA release from reactive astrocytes leads a complete restoration of impaired synaptic plasticity, learning and memory. Furthermore, it has been reported that GABA and MAOB are significantly upregulated in astrocytes of post-mortem AD brains [82]. In recent years, MAOB has emerged as a potential therapeutic target for AD treatment and has been studied with selegiline, an irreversible MAOB inhibitor [117,118]. While treatment with selegiline improved short-term cognitive function [119], it has also been reported that there were no significant long-term effects [82]. A subsequent study also found that diamine oxidase (DAO) activity increases following MAOB inhibition and the production of GABA is sustained in spite of long-term MAOB inhibition [109]. Taken together, the dysregulation of GABA homeostasis by astrocytes appears to play a crucial role in the pathogenesis of AD.

Attention-Deficit/Hyperactivity Disorder (ADHD) is a neurodevelopmental or behavioral disorder characterized by symptoms such as hyperactivity, impulsivity, and inattention. In the cerebellum from G protein-coupled receptor kinase-interacting protein-1 (GIP1) knockout mice, which is an ADHD model [120], decreases in glial GABA immunoreactivity and attenuation of the granule cell tonic GABA current were reported [121]. The authors suggest that this attenuation increases the excitatory/inhibitory synaptic input ratio and might be responsible for hyperactivity in ADHD. However, although GIT1 KO mice have decreased vGAT and GAT67 at inhibitory presynaptic terminals to hippocampal CA1 pyramidal cells [120], the mechanism of reduced glial GABA in the cerebellum remains unknown.

Finally, mutations in the gene *BEST1* are associated with five clinically distinct retinal degenerative diseases, which are referred to as "bestrophinopathies", and are: Best vitelliform macular dystrophy, autosomal recessive bestrophinopathy, adult-onset vitelliform macular dystrophy, autosomal dominant vitreoretinochoroidopathy, and retinitis pigmentosa [122]. Despite the discovery of the gene *BEST1* in 1998 [123,124], the mechanisms of pathogenesis underlying retinal degeneration by the *BEST1* mutations remain unknown [77,122]. In contrast, decreased the amplitude of the tonic GABA current in cerebellar granule cells and improved motor function have also been reported in Best1 knockout animals [84]. However, so far, no evidence has been reported showing a clear relationship between central nervous system diseases (except retina) and functional abnormalities of Best-1 expressed in astrocytes.

Thus, it has been reported that the astrocyte-dependent tonic GABA current is involved in the pathophysiology of various neurological diseases. However, despite the numerous additional reports of changes in tonic GABA current, further investigation is necessary to determine whether these changes result from neural or astrocyte activity.

## 6. Conclusions

Since their discovery, astrocytes have been thought of as passive supporting cells for neurons in the central nervous system. More recent results from many studies indicate that astrocytes play an active role in generating and propagating Ca^2+^ waves and are key regulators of chemical synaptic transmission. In this review, we have focused on how astrocytes can shape signal transmission at inhibitory synapses, and the pathophysiological consequences of disturbing these functions. While substantial attention has been focused on the role of astrocytes in excitatory synaptic transmission, the findings described in this review indicate that astrocytes are likely to play profound roles at inhibitory synapses. Thus, future studies clarifying the mechanisms underlying astrocyte actions at inhibitory synapses are warranted and likely to provide important opportunities to develop novel treatments for a range of currently intractable neurological diseases including epilepsy.

## Figures and Tables

**Figure 1 ijms-20-02964-f001:**
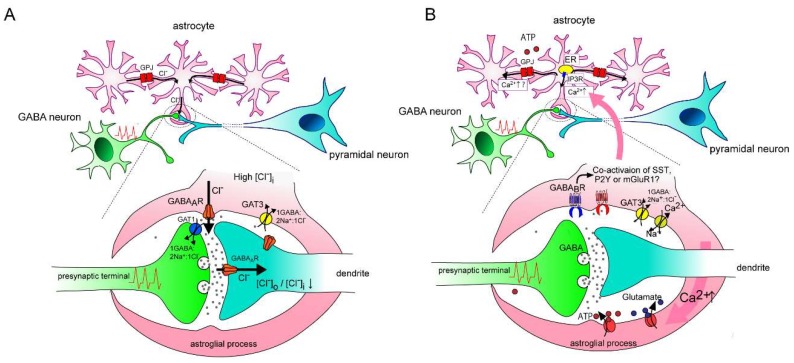
Hypothetical schema depicting the contribution of an astrocytic GABA_A_ receptor (GABA_A_ R) on the modification of GABAergic transmission. (**A**): Under high [Cl^–^]i in astrocytes, spillover of GABA evoked by repetitive interneuron firing induces Cl^–^ efflux via astrocytic GABA_A_ receptors, which are localized near the synaptic cleft. The relatively distal localization of GAT3 mediates Cl^–^ influx in cooperation with GABA uptake, but its contribution may be negligible because the diffusion of GABA is limited by neuronal GAT1. Astrocytic gap junctions (GPJ) tightly coordinate [Cl^–^]i within coupled astrocytes. GABA_A_R-induced Cl^–^ efflux causes a siphon effect that induces simultaneous compensation by Cl^–^ influx from non-activated astrocytes via GPJ, so that the driving force for Cl^–^ efflux by astrocytic GABA_A_R is maintained. This astrocytic GABA-mediated Cl^–^ efflux might help maintain the postsynaptic transmembrane Cl^–^ gradients of GABAergic synapses by spatially buffering [Cl^–^]o. (**B**): A result similar to (**A**) is observed on Ca^2+^ increase via GABAergic signaling in astrocytes. An astrocytic GABA_B_ receptor senses GABAergic signaling and its activation induces intracellular Ca^2+^ increase from the endoplasmic reticulum (ER) via IP3 receptors. Co-activation of other Gq type metabotropic receptors might be required for activating the IP3 receptor. Na^+^ influx is accompanied by GABA transportation via GATs (GAT3 is dominant in the majority of astrocytes), which results in Ca^2+^ influx via a Na^+^/Ca^2+^ exchanger in reverse mode. Ca^2+^ rise triggers release of gliotransmitters including ATP and glutamate, which modulate both presynaptic and postsynaptic functions variably. Ca^2+^ increase can propagate to neighboring astrocytes via gap junctions and/or ATP release, which may allow the specially synchronized cording of astrocytes.

**Table 1 ijms-20-02964-t001:** Subtype names of GABA transporter (GAT) across species.

Species	Gene			
	*SLC6A1*	*SLC6A12*	*SLC6A13*	*SLC6A11*
Mouse	GAT1	GAT2	GAT3	GAT4
Rat, Human	GAT1	BGT1	GAT2	GAT3

(BGT1, betaine–GABA transporter).

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
