# Peer review of "Diverse Actions of Astrocytes in GABAergic Signaling"

_ijms, 2019, doi:10.3390/ijms20122964_

Reviewer 1 Report

The review by Ishibashi et al., “Diverse actions of astrocytes in GABAergic signaling” attempts to focus on the astrocytic GABAergic neurotransmission.
This manuscript, however, is not clearly enough written so that it is understandable to the readers. The writing often lacks clarity and sharpness,
and several sections are poorly organized. The authors should correct carefully on matters of the organization and contents throughout the paper. In addition, the English is substandard, interfering with the readability of the paper.

1.  First part of Abstract or of Introduction is redundant.
2.  Section 2-1 first part. Is there specific subunit pattern of GABA A  receptor in astrocyte?  The  description  of  “…exclusive  expression  of  Γ1  and  Γ2  in astrocytes…” and “…the lack of Γ2 subunit in astrocytes…” is confused.
3.  Section 2-1 middle part. What  is “type 2 GABAergic synapses on Purkinje cells”? The  images  of  the  synapse  contrasting  type  1  GABAergic synapse may be helpful to the readers.
4.  Section  2-1  last  part.  I  could  not  understand  the  description  regarding  the collapse of neuronal [Cl - ]o/[Cl - ]I gradient.
5.  Section  2-2.  The  content  of  GABA B   receptor-mediated  astrocytic  Ca 2+  signaling should be concise focusing on more important points. The same is true of section 2-1.
6.  Section 3-1, GAT subtype. “Mouse GAT4 is named GAT-3 in humans” while in section 2-3, “GAT3 is assumed to be expressed exclusively in astrocytes”. Is GAT-3 and GAT3 the same or different transporter?
7.  Section  3-1  middle  part.  I  could  not  understand  following  sentences.  “…in immunocytochemical  verification,  it  has  been  reported  that  GAT3  is expressed  at  high  levels  in  the  astroglial  process  of  neuropil  close  to  cell bodies and dendrites containing symmetrical and asymmetric synapses.”
8.  It  would  be  better  to  combine  Section  3-2  and  Section  3-3  as  other mechanisms for GABA release from astrocytes.
9.  Section 3-4 including GABA metabolism should be removed from Section 3 “GABA transport systems in astrocyte”.
10. Section  4,  Pathophysiological  impacts.  There  is  no  description  of  Cl -  dyshomeostasis  in  relation  to  the  pathogenesis  of  neurological  disorders while in the manuscript mentioning “…we have focused on the relationship between altered astrocytes and Cl -  or GABA homeostasis…”.  

Author Response

We thank the reviewers for their careful reading and helpful comments.  We believe we have addressed all of the issues raised and feel the MS is now suitable for publication. Our responses are in text following each reviewer’s comments below.

The review by Ishibashi et al., “Diverse actions of astrocytes in GABAergic signaling” attempts to focus on the astrocytic GABAergic neurotransmission. This manuscript, however, is not clearly enough written so that it is understandable to the readers. The writing often lacks clarity and sharpness, and several sections are poorly organized. The authors should correct carefully on matters of the organization and contents throughout the paper. In addition, the English is substandard, interfering with the readability of the paper.

Response: Sorry for the lack of clarity. In the revised manuscript we have reduced possible redundancies, clarified text identified as unclear, and improved the English.  We have also worked to provide a more focused and streamlined organization to the MS.

1.First part of Abstract or of Introduction is redundant.

Response: We have streamlined the abstract and Intro, and sought to remove redundancies. We believe it is now clearer and better sets out our goals for the review.

2.Section 2-1 first part. Is there specific subunit pattern of GABAA receptor in astrocyte? The description of “…exclusive expression of Γ1 and Γ2 in astrocytes…” and “…the lack of Γ2 subunit in astrocytes…” is confused.

Response: Yes.  As we now more clearly indicate in section 2-1, RT-PCR and immuno-EM methods identified the presence of gamma 1 and 3 subunits and the absence of gamma 2 subunits in astrocytes.  We also indicate “the ρ subunit is expressed in astrocytes of various brain regions…”

3.Section 2-1 middle part. What is “type 2 GABAergic synapses on Purkinje cells”? The images of the synapse contrasting type 1 GABAergic synapse may be helpful to the readers.

Response: “type 2” refers to the classical Gray’s type 2 or “symmetric” synapse ultrastructure typically associated with functionally inhibitory synapses (see: https://synapseweb.clm.utexas.edu/type-2-synapse).  This is a basic concept and a picture is not necessary.  For clarity in the MS we now refer to them as “(type II, symmetric)”

4.Section 2-1 last part. I could not understand the description regarding the collapse of neuronal [Cl-]o/[Cl-]i gradient.

Response: We have made this clearer by indicating that Cl- accumulates inside of the cell thereby reducing (collapsing) the transmembrane gradient.

5.Section 2-2. The content of GABAB receptor-mediated astrocytic Ca2+ signaling should be concise focusing on more important points. The same is true of section 2-1.

Response: We edited both sections to improve the English and to make the writing more concise. Of course, we have tried to focus on the most important points but without more specific guidance, it is not clear what was perceived as unimportant.  We believe we have captured the most important points relevant to these topics and have now made the text more readable.

6.Section 3-1, GAT subtype. “Mouse GAT4 is named GAT-3 in humans” while in section 2-3, “GAT3 is assumed to be expressed exclusively in astrocytes”. Is GAT-3 and GAT3 the same or different transporter?

Response: Because of the naming practices in the field, we recognize this is very confusing.  We have now revised the text and added a table to be explicit about the names of mouse and human GAT genes. We feel this has made this section much more understandable.

7.Section 3-1 middle part. I could not understand following sentences. “…in immunocytochemical verification, it has been reported that GAT3 is expressed at high levels in the astroglial process of neuropil close to cell bodies and dendrites containing symmetrical and asymmetric synapses.”

Response: We have reworded the sentence for clarity to read: “Furthermore, it has been reported by immunocytochemical localization that GAT3 is expressed at high levels in astroglial processes near symmetric and asymmetric synapses [65].

8.It would be better to combine Section 3-2 and Section 3-3 as other mechanisms for GABA release from astrocytes.

Response: Changed as suggested.

9.Section 3-4 including GABA metabolism should be removed from Section 3 “GABA transport systems in astrocyte”.

Response: As suggested, we made a new Section 4 from 3-4.

10.Section 4, Pathophysiological impacts. There is no description of Cl- dyshomeostasis in relation to the pathogenesis of neurological disorders while in the manuscript mentioning “…we have focused on the relationship between altered astrocytes and Cl- or GABA homeostasis…”.

Response: Good point.  This is now Section 5 and we have removed “Cl− or” from the sentence.

Reviewer 2 Report

This review article sheds light on the role of astrocytes in GABA-mediated signaling at inhibitory tripartite synapses. In the first half of the review, the authors discuss GABAA and GABAB receptor mediated signaling from neurons to astrocytes. Next, studies on different GABA transporters expressed in the astrocytes in different parts of the brain are discussed. Additionally, the authors discuss non-traditional channels for GABA transport, i.e. anion channels and gap junction hemichannels. Finally, the authors discuss in short alterations in GABA/ Cl- homeostasis in astrocytes in different disease conditions.

The strengths of the article include well organized, structured, in depth review along with appropriate references.

My minor comments are;

Introduction, para 2: “GFAP is a widely used marker for astrocytes”; provide a reference here.

Introduction, para 3: “At synapse that we focus on this review…” This sentence does not make sense and needs to explained.

Pathophysiological impacts: “In cerebellar from G protein coupled receptor….”; change cerebellar to cerebellum.

Author Response

We thank the reviewers for their careful reading and helpful comments.  We believe we have addressed all of the issues raised and feel the MS is now suitable for publication. Our responses are in text following each reviewer’s comments below.

Introduction, para 2: “GFAP is a widely used marker for astrocytes”; provide a reference here.

Response: we added an appropriate reference. This reference make mention of GFAP as astrocyte marker with other markers. For example, S100B, vimentin, Aldh1l1.

Introduction, para 3: “At synapse that we focus on this review…” This sentence does not make sense and needs to explained.

Response: We removed the sentence and edited the Intro for English and clarity.

Pathophysiological impacts: “In cerebellar from G protein coupled receptor….”; change cerebellar to cerebellum.

Response: Fixed.

Round  2

Reviewer 1 Report

The authors have responded in a satisfactory manner to our multiple points of criticism. I recommend that the review be published as it now stands.